# Self-Associating Polymers Chitosan and Hyaluronan for Constructing Composite Membranes as Skin-Wound Dressings Carrying Therapeutics

**DOI:** 10.3390/molecules26092535

**Published:** 2021-04-26

**Authors:** Katarína Valachová, Ladislav Šoltés

**Affiliations:** Centre of Experimental Medicine, Institute of Experimental Pharmacology and Toxicology, SAS, Dúbravská cesta 9, SK-84104 Bratislava, Slovakia; ladislav.soltes@savba.sk

**Keywords:** antioxidants, chitin, hyaluronic acid, l-(+)-ergothioneine, MitoQ, resveratrol, SkQ, wound healing

## Abstract

Chitosan, industrially acquired by the alkaline *N*-deacetylation of chitin, belongs to β-*N*-acetyl-glucosamine polymers. Another β-polymer is hyaluronan. Chitosan, a biodegradable, non-toxic, bacteriostatic, and fungistatic biopolymer, has numerous applications in medicine. Hyaluronan, one of the major structural components of the extracellular matrix in vertebrate tissues, is broadly exploited in medicine as well. This review summarizes that these two biopolymers have a mutual impact on skin wound healing as skin wound dressings and carriers of remedies.

## 1. Introductory Remarks

### 1.1. Pharmacokinetics

The drug can enter the body in several ways depending on the route of administration, which can be parenteral, or enteral, i.e., into the gastrointestinal tract. The concentration of the drug at the site of its action depends on several processes, which include drug absorption (resorption), distribution in individual tissues, and elimination. The principal goal of drug administration in any form is the entry of the drug to the bloodstream through circulation by which the drug is distributed to various parts of the body. On the other hand, to transfer the drug to the site of action or to be metabolized and excreted from the body, the drug must be transported across cell membranes. This transport is passive or active, i.e., mediated by carriers.

Concerning lipophilic drugs, a passive transport through the lipid bilayer of cells occurs by simple diffusion: the rate of diffusion depends on the size of the drug molecule, the thickness of the membrane and the size of the resorption area. Drugs that are weak acids or bases in the polar milieu exist simultaneously as ionized and non-ionized fractions. The non-ionized drug form is usually liposoluble and, therefore, readily crosses membranes. The ionized fraction is hydrosoluble and is difficult to pass through the lipid bilayer. The ratio of both drug fractions is determined by the degree of ionization of a weak acid—p*K*_a_—(p*K*_b_ for the weak base), which is a physicochemical parameter: if the p*K*_a_ of the drug is identical to the pH of the solution, in which the drug is dissolved, then 50% of the drug is ionized and 50% remains in a non-ionized form.

A wide range of drug dosage forms can be ingested orally and varies from liquid (solutions) and semi-solid forms (emulsion pastes) to solid ones (tablets, capsules, granules, powders). The disadvantage of oral ingestion is a slower onset of action of the drug, its uneven absorption, degradation of the drug in the stomach, or interaction with food. The so-called first pass effect is the uptake of a fraction of the ingested drug by the liver, which ultimately reduces the dose of the drug that enters the bloodstream. One of the modes of drug applications to circumvent the adverse first pass effect is to administer the drug through the skin, mucosa. This route of administration is primarily accompanied with a local/pharmacological effect of the remedy. Since the skin is a relatively thick membrane, liposoluble drugs readily pass through the membrane. In contrast, water-soluble drugs penetrate to skin slowly and in small amounts.

When sustain- or smart-release of a drug or of an active/medical principle is incorporated into a reservoir/carrier, e.g., into a composite membrane, it is necessary to replace the membrane with a new one, e.g., every day. The initial flow of the drug as well as the final flow, i.e., 24 h after the application of the composite membrane must correspond to the drug uptake into the skin in a concentration, which will be effective in the site of the drug action—see Scheme 1.

Transdermal drug delivery systems (TDDS) are formulations containing one or more drugs that are applied to the skin to achieve a systemic effect [1,2,3,4,5,6,7,8]. The first experiments with TDDS contained immunomodulators, contraceptives, and also antibiotics. The latter, especially for military purposes, date back to the 1970s. Currently, drugs incorporated into TDDS must have several characteristics, namely (i) low therapeutic level, (ii) good transport across the skin barrier, (iii) the affinity of the drug to the skin must be higher than to the delivery system, (iv) the molar mass range of the drug should be 100–800 Da, (v) the solubility of the drug in water and in octanol must be over 1 mg/mL, and (vi) the drug must have a minimum of polar/zwitterionic functional groups. Further criteria are: insignificant first pass metabolism of the drug in the skin, therapeutic efficacy at a dose <10 mg/day, biological half-life of the drug 6–8 h, as well as minimal toxicity and allergenicity to the skin during long-term application. The patient’s preferences for the use of TDDS are in particular the improved patient compliance due to simplicity of application. The patch size limit is 50 cm^2^, with the maximum transdermal dose of the drug being limited to 5–20 mg/day [9]. In the last years, the production and application of TDDSs have been broadly investigated in the field of drug sustain delivery, along with drugs in the form of conjugates with a polymer and/or hydrogel depot systems [10].

It is of a great interest in the scientific community to investigate the alternatives of how to administer drugs/biologically active substances to organisms safer. Namely, the mode to administer the substance into the body by a more or less controlled penetration through the skin has been accompanied with a valuable number of easy-to-implement procedures for the incorporation of several types of exogenous substances into a carrier/TDDS [11]. Achievements in widespread use of TDDS (patches) are documented by data in Table 1.

### 1.2. Tissues Covering the Organisms

Multicellular organisms, whose body forms an integrated set of organs, are usually separated from the surrounding environment by a soft, flexible tissue covering their body—the skin or by a hard cover, e.g., the exoskeleton of crustaceans and mollusks. One of the hard exoskeleton biopolymeric components is chitin. The elastic skin besides numerous components is also composed of another biopolymer, namely hyaluronan (HA).

The skin in an adult human covers an area of approx. 1.5–2 m^2^; its thickness is in a range of 0.5–5 mm. The slightly acidic pH value of 4.5–5.5 ensures that the skin surface prevents the growth of bacteria, yeasts, and fungi. Skin has several functions such as protective, thermoregulatory, sensory, and immune. Of the protective functions, we denote, e.g., protection against adverse external physical (injury, sunlight), chemical and biological (invasion of microorganisms) properties. Several minor mechanical injuries and skin damages heal spontaneously. However, the more severe skin defects in patients may be accompanied with chronic injuries with a fatal outcome.

### 1.3. Chitin, Chitosan, and Hyaluronan

The structural unit of the chitin biopolymer—{C_8_H_13_O_5_N}_n_—*N*-acetyl-d-glucosamine (GlcpNAc), or more precisely 2-(acetylamino)-2-deoxy-d-glucose, the monomer (*n* = 1) is well water soluble. The crystalline structure of chitin is the result of a complex biosynthesis, which is accompanied by a reorganization of the chains to form antiparallel arrangements in case of α-chitin or parallel in case of β-chitin. These features explain their insolubility and different reactivities. Chitin in a solid phase is used in industry in many processes, e.g., nanocrystal chitin particles serve as stabilizers of food hydrocolloids [41]. There is a growing interest of industry in products acquired from chitosan. To a much greater extent, a chitin derivative—chitosan—is employed as scaffolds in studies concerning tissue growing and wound healing.

Chitosan, a linear polysaccharide composed of randomly distributed GlcpNAc and its deacetylated unit β-(1→4)-linked D-glucosamine, is synthesized by partial deacetylation of chitin with, e.g., aqueous NaOH solution. The deacetylation degree in commercially available chitosans varies from 60 to 100%. The average molar mass of commercially used chitosan macromolecules is up to 20 kDaltons. The amino groups in chitosan have p*K*_a_ values of approx. 6.5, thus, in neutral aqueous solution (pH 7) the macromolecules are positively charged.

Hyaluronan (HA), a biopolymer, is composed of repeating disaccharide units of GlcpNAc and d-glucuronic acid, linked exclusively by β-linkages. HA is a member of the glycosaminoglycan family [42,43]. Commercially used HAs have a wide interval of mean molar masses reaching up to several megadaltons (MDa). Due to the presence of free d-glucuronic acid groups, HAs in aqueous solutions behave like polyanions: the overall p*K*_a_ values of HA carboxyl groups are approximately in the range 3–4 [10,44]. HA is a major component of epidermis and dermis of the skin. HA is broadly used as an excellent moisturizer in cosmetics and skin-care products. The negative charge of HA macromolecules and the acidic pH of the skin surface are the main factors for their good mutual compatibility.

### 1.4. Chitosan-Hyaluronan Associates

While chitosan itself forms a perfect film after drying of its slightly acidic solutions (usually in 2% aqueous acetic acid), HA lacks such film forming propensities. Thus, when combining the two solutions in appropriate ratios, one may obtain a viscous solution (see Figure 1), which after drying easily forms a thin film with much higher tear resistance than a film formed only from the chitosan solution. After the addition of another component such as an “active principle” to the pre-formulated viscous solution of those two polymers, we prepare a composite membrane, which is ready-to-use for treatment of, e.g., difficult-to-heal chronic skin wounds.

The coacervate/polyionic complex hydrogel represents a polar/aqueous milieu, however, when the active principles are poorly water soluble, they are dissolved in, e.g., dimethylformamide or dimethylsulfoxide. Another, more progressive approach, includes the incorporation of a liposoluble drug into either a liposome or into the cavity of a cyclodextrin derivative; thereby, we obtain a readily water soluble drug form [45]. Further additives during preparation of the polyionic hydrogel can be bacteria- and/or fungistatics (e.g., water soluble NaN_3_, hydrophobic thymol, or colloidal nanosilver).

A minor drawback, however, exists when the membrane is prepared by mixing the two polymer solutions. While HA is readily water soluble, chitosan needs to be dissolved in a diluted acid (most often 2% aqueous acetic acid). To eliminate the traces of CH_3_COOH, an additional step such as adding of the membrane to an alkaline solution (usually in aqueous NaOH solution) is recommended. To solve this limitation, another investigation of the scientists has been focused on preparing water-soluble chitosan derivatives [46,47,48,49,50].

Physicochemical crosslinking of the positively charged chitosan chains with biopolymeric negatively charged HA networks can result in preparation of novel advantage materials applicable in medicine. Such composites can serve as artificial skin-wound dressings, which often carry some therapeutics, and/or some other bioactive components [51].

Yet, some comments are necessary to add: chitosan itself and its water-soluble derivatives form easily an elastic foil, which can be used as wound dressings and TDDSs [52]. Depending on the foil thickness, an additional benefit might be a good transparency of plastic sheets. Generally, wound-covering foils/sheets prepared this way have some advantageous biological properties, e.g., antimicrobial [53]. To increase the chitosan-derived-foil resistance to ruptures, several low and high-molar-mass cross-linkers including HA were employed [54,55]. Some people, however, might demonstrate an allergic response to products derived from either crustaceans and mollusks or a HA cross-linker. Thus, chitosan-derived-foils are not suitable for patients with a previously diagnosed allergy to those materials. In such a situation, an anaphylactic shock can happen, where epinephrine injection is the first-line treatment.

One of the most promising alternatives is the preparation of composite membranes, which the drug/an active principle is smartly released from. The drug/the active principle should penetrate into the skin, and be absorbed to the bloodstream; thereby, the drug is distributed to the site of its action.

In this communication, we summarize the achievements underwent in our laboratory and simultaneously in some laboratories worldwide.

Some drugs/medical principles incorporated to membranes of chitosan and HA are summarized in Table 2.

### 1.5. Treatment of Difficult-to-Healing Skin Wounds

The body’s response to tissue injury in a healthy individual is a sequential physiologic process that results in re-epithelialization, resolution of drainage, and recovery of function of the affected tissue. Chronic wounds, however, do not follow this sequence of events. For example, tissues of leg and decubitus ulcers or burns of higher degree are chronically inflamed [3,16]. Besides necrotic cells, such tissues contain other cells, in which oxygen uptake is insufficient due to a damage of extracellular glycocalyx (state of hypoxia). It is well known that mitochondria in cells enhance the production of reactive oxygen species (ROS) during hypoxia. ROS such as O_2_^●−^, H_2_O_2_, and of those forming ^●^OH radicals are the compounds that can cause destruction of the tissue. It is generally recognized that ROS originate in the elevated amounts from the stressed cells. One way to effectively reduce an adverse flow of ROS from mitochondria is to use the so-called mitochondrially-targeted antioxidants (MTAs). MTAs can selectively enter mitochondria in cells. MTAs with a positively charged carrier of the “Skulatchev ions” type are already used commercially. MTAs have to be applied on the site of an inflamed tissue at a relatively low concentration during a prolonged period of time. The authors of this paper suggest that MTA incorporated into membranes formed by two biopolymers: high molar mass hyaluronan (HMM HA) and chitosan could reduce or scavenge the flux of ROS damaging skin tissues. An appropriate combination of HA and chitosan allows the formation of very stable biofilms with a certain excess of the negative charge. Incorporation of MTA into such a biofilm results in formation of biomembranes [75], whereas MTA is released gradually during an extended period. One of the especially important advantages of the aforementioned combination is that MTA in an oxidized form can be preserved for an illimitable long period, because MTA is not oxidized spontaneously. MTA is activated, i.e., reduced after entering the cell. It is possible to maintain membranes in a mildly humid milieu in an appropriate bag (pocket) and to apply them on a wound at any time when needed. Since after a certain time MTA penetrates from the membrane and is incorporated into the wound tissue, the membrane can be readily removed and substituted by a new one. Incorporation of the cytoprotectant into such a membrane results in the formation of composite membranes, whereas the cytoprotectant is smart-released during an extended period [76]. Figure 2 displays wound healing of ulcers on a man’s leg. The achieved progress during the periodical application (once in three days) of a properly designed composite membrane led to patenting the novelty and originality of the approach applied [75,76].

By varying the ratio of the polymers chitosan and HA it is easy to adjust the proper resulting charge of the storage space/reservoir of the drug—the active medical principle, which is partially cationic or anionic.

Since the molar mass of disaccharide unit of completely deacetylated chitosan equals to the approximate molar mass of HA disaccharide, the ratio 1:1 *w/w* should result in nil reservoir charge. To obtain identical zero charge when using 50% deacetylated chitosan, it is necessary to blend two parts of chitosan with one part of HA. Thus, e.g., due to a little surplus of positive reservoir polarity, the negatively charged drug can be gradually released in accord to concentration gradient between the membrane surface and the site of drug action and on mutual electric attraction between the membrane and the active medical principle. In the opposite situation, when polarity of the reservoir is negative and the drug has a positive charge, the drug gradually releases based on the above-specified conditions. Yet, it should be pointed out that when the drug is water soluble, it is appropriate to harmonize the charges of both the reservoir and the drug to identical ones, either positive or negative. Although, the treatment with a loosely trapped drug is the primary aim, it is necessary to mention that the membrane fabricated from chitosan and HA is frequently applied as artificial skin, which promotes the treatment of superficial chronic tissues damages and/or difficult-to-heal wounds [67,68,69,70,76,77].

### 1.6. Drug Release from Chitosan-HA Membrane

Some additional surmountable obstacles should be mentioned here. When one would like to prepare a composite membrane composed of chitosan and HA with a charged/hydrophilic compound of low-molar-mass, it is necessary to keep in mind that the release of a hydrophilic low-molar-mass compound from a hydrophobic depot runs well; however, the electrically neutral hydrophilic compound does not release from the hydrophilic reservoir (cf. Scheme 2). On the other hand, the charge on the drug molecule, especially that of plus in chitosan-HA membranes, characterized by positive charge originated from chitosan, resulted in a real incompatibility between the reservoir and the drug. In such a case, the drug is released from the mass of the reservoir to its surface.

The positively charged drug, present in excess on the surface of the positively charged composite membrane, is immediately disposable for either entering the cells of the skin or for elimination from epidermis or dermis to systemic blood circulation. Thus, cationic active/medical principles or drugs “fired” from positively charged reservoirs are practically not tightly trapped by its carrier.

### 1.7. Chitosan-HA-MTA Composite Membranes

As already reported in the paragraph “Treatment of difficult-to-healing skin wound”, numerous mitochondrially-targeted molecules have been already prepared [78]. Of those two, representatives such as MitoQ and SkQ should be mentioned here (cf. Table 2). Any positively charged MTA must first be transported across the cell membrane, which electrical potential (Δψ) ranges from −30 to −60 mV. After penetrating into the cell, the MTA molecule is transported across the charged bilayer barrier (Δψ = −150 to −180 mV) surrounding the mitochondrion. The primary function of the MTA molecules reaching the internal space of mitochondria is to convert the undesirable surplus of O_2_^●−^ anion radicals to molecules of dioxygen (O_2_). Due to the oxidative reaction (2O_2_^●−^→2O_2_ + 2 electrons), the two electrons are trapped by the molecule of MitoQ (mitoquinone) or SkQ yielding the molecule of mitoquinol or SkQ1 (cf. Scheme 3 [79,80]).

The rarely claimed observation concerning the application of, e.g., MitoQ is its very narrow therapeutic window. It means that when the dose of this mitochondrially-targeted antioxidant exceeds an optimal level, it could be counterproductive [81,82,83,84]. Yet, when the appropriate dose of MitoQ is incorporated into the composite membranes skin wounds healed effectively [61]. Because of the necessity to control the SkQ dose due to its very narrow therapeutic window, the Russian investigators applicated this MTA exclusively within, e.g., Visomitin SkQ1—eye drops treating cataract [85]. One drop per eye three times/day means for a patient to administer approximately 14 ng of SkQ1. SkQ1 is a prodrug, which when entered the mitochondria is converted to its active form and, thus, can scavenge noxious superoxide anion radicals. Analogously, as we observed [61], the MTAs incorporated into chitosan-HA membranes could have appropriate applications, even like a TDDS tool.

### 1.8. Chitosan-HA-Antioxidant Composite Membranes

One of the functionally essential components of skin is HA. HMM HA, which has viscous properties, is located around the skin cells and forms the so-called extracellular matrix. HMM HA in skin is sensitive to the attack by free radicals. The four steps of perpetual radical HA degradation are: initiation, propagation, transfer, and termination (see Scheme 4).

According to the cascade of reactions showed in Scheme 4 it is obvious, that to interrupt the perpetual free-radical HA degradation [cf. reaction “intermediate free-radical products” in Scheme b, c, e, f we must apply an antioxidant, which acts as H atom donor. Such a property can be attributed to l-(+)-ergothioneine and resveratrol (cf. Table 2).

### 1.9. Chitosan-HA-Resveratrol Composite Membrane

Trans-3,5,4′-trihydroxystilbene (C_14_H_12_O_3_) exists as two isomers, namely, *cis-*(*Z*)- and *trans*-(*E*)-resveratrol, which belong to phytoalexins—the compounds of several plants produced as a response to injury or when the plant is attacked by pathogens. Currently, *trans*-resveratrol is spread among the human population as a dietary supplement, however, there is no significant evidence that this substance improves lifespan or has a substantial effect on any human disease [86]. The solubility of resveratrol in water equals approximately 0.05 mg/mL and this compound belongs to antioxidants. Resveratrol is extensively extruded from any hydrophilic, e.g., chitosan-HA membrane. Resveratrol, owing to its lipophilic nature, penetrates easily into skin where its molecule with two diols in meta position might first undergo isomerization, which yields an intermediate compound bearing two diols in the ortho-position. The intermediate during the next oxidation-reaction-step converts to an ortho quinine type product (cf. the upper panel in Scheme 5). The reaction yield (2 electrons plus 2 H^+^) is sometimes classified as a transfer of two H atoms [87]. Thus, to date trans-resveratrol is designated as an efficient scavenger of hydroxyl and hydroperoxyl (^●^OOH) radicals [88]. As reported by Shang et al. [87], the compounds within the top panel could, however, dimerize by the Diels–Alder condensation reaction. Potentially, another reaction flow chart could be proposed as shown in Scheme 5, the bottom panel. By such an electron transfer conjugation reaction another quinine compound could be formed. Both reaction products represented in Scheme 5 (cf. upper and lower panels) are, however, highly reactive counter partners to, e.g., condensation with endogenous aliphatic amines via a Schiff-base reaction. That is why a great caution has been claimed with uncontrolled use of polyphenols.

### 1.10. Chitosan-HA-Ergothioneine Composite Membrane

l-(+)-Ergothioneine (see Scheme 6) is a sulfur-containing derivative of the amino acid histidine. l-(+)-Ergothioneine, synonymum (S)-α-carboxy-N,N,N-trimethyl-2-mercapto-1H-imidazole-4-ethanaminium, C_9_H_15_N_3_O_2_S, molar mass 229.3 g/mol is a non-toxic bioactive molecule. l-(+)-Ergothioneine is tautomeric and in neutral aqueous solutions exists predominantly in the thione form, which may account for l-(+)-ergothioneine’s resistance to autoxidation. l-(+)-Ergothioneine’s sulfhydryl group in its thiol form highlights its antioxidant properties such as two H atoms donoring property.

Analogously to *trans*-resveratrol, l-(+)-ergothioneine is consumed by human beings as a dietary supplement. Contrary to slight solubility of resveratrol in water, l-(+)-ergothioneine forms aqueous solutions easily and is scarcely extruded from a hydrophilic reservoir. Thus, the amount of the polycationic chitosan should be properly set, which results in forming a membrane, whose charge supports the flux of l-(+)-ergothioneine from the reservoir composed of chitosan and HA. The positive charge of l-(+)-ergothioneine molecules allows their penetration into the skin and after reaching the epidermis also the drug absorption into the blood stream. The presence of an endogenous protein such as a high affinity transporter OCTN1 of l-(+)-ergothioneine allows the substance efficient distribution in the body. One can easily anticipate that l-(+)-ergothioneine molecules in thiol forms are appropriate donors of H atom and should scavenge not only –CO^●^ and –COO^●^ type radicals (cf. Scheme 4c,e,f) of the perpetual free-radical degradation of HA within the inflamed skin, but also further ROS generated in the injured body tissues distant from the site of application of the composite membrane.

### 1.11. Limitations/Dangers of Not Critical Application of Antioxidants

Either MTA or any other orally ingested antioxidant along with its potentially positive action in parallel could cause a large adverse outcome, namely that of a reductive stress. Thus, here we claim: at present, however, a great boom in various fields of industry is to market numerous compounds, simply denoted as antioxidants with uncritical glorification of protective effects of these compounds prevailingly against the so-called oxidative stress. Yet, a poorly informed consumer following the “law of mass action” (more is better) could overload his body with compounds/antioxidants, which can result in a high redox disbalance, thereby to reductive stress. Figure 3 illustrates such circumstances:

To comment the above mentioned Figure 3, as anticipated by Xiao and Loscalzo [89], one can state that under physiological conditions, cellular redox buffers in a healthy subject have sufficient capacity (termed basal redox buffer capacity (ReBC)) to maintain cellular oxidants and reductants/antioxidants at physiological levels. When cells are subjected to oxidative or reductive insults, redox buffers increase to a certain level (termed compensatory ReBC) to counteract these redox stresses and restore redox homeostasis. Under these circumstances, cellular oxidants and reductants are still maintained within physiological ranges. However, when this compensatory response reaches a maximum, the ReBC is exceeded and oxidative or reductive stress occurs. Importantly, reductive stress diminishes cellular reactive oxygen species (ROS) levels at physiological levels and, thus, perturbs their signaling functions. From a different viewpoint, reductive stress can also promote ROS production (e.g., by partially reducing oxygen) and, thus, is proposed to promote oxidative stress in essence, depending on the redox couples in which these ROS are engaged.

## 2. Addendum

The informed readers of this journal probably would welcome the additional information on the molecule of the antioxidant—ergothioneine, which follows here:

In the period of origin of animals, 2–3 billion years ago, the Earth was not protected by an atmosphere containing oxygen. Cosmic radiation decomposed molecules of water and the forming radicals (particularly ^●^OH) were responsible for: (i) mutation of organisms and (ii) their destruction/death. However, how to provide the survival of organisms affected by “positive mutations”? In this prehistoric time, there existed such organisms, which were capable of scavenging ^●^OH radicals; thereby, these organisms prolonged their existence in a non-mutated form. The compound discovered by these organisms is denoted as l-ergothioneine [90].

In the last years, scientists have discovered that, surprisingly, some tissues in humans and animals as well contain unexplainably high levels of l-ergothioneine, and since there is an unequal distribution of l-ergothioneine in the organism for some reason, the research was focused on finding a selective/specific l-ergothioneine transporter. *Et voilà*, a transporter was found. It is a protein—a high affinity OCTN1 vehicle [91]. Further research showed that the synthesis of OCTN1 protein is encoded in the human DNA; yet, l-ergothioneine is for humans recognized as a foreign compound. Thus, the challenge was:to solve a patentable procedure of l-ergothioneine synthesis [92];to persuade consumers to buy l-ergothioneine and to ingest it as a natural supplement.

In accord with Paul and Snyder [93], one could say that although l-(+)-ergothioneine was isolated a century ago, its physiologic function has not been so far clearly established. Because of its dietary origin, l-(+)-ergothioneine represents a vitamin, whose physiologic roles include cytoprotection. The only organisms known to synthesize l-(+)-ergothioneine are bacteria belonging to the genus Actinomycetales (example mycobacteria) and non-yeast like fungi, which include members of the division Basidiomycota and Ascomycota. These microbes synthesize l-(+)-ergothioneine from l-histidine via an intermediate hercynine—a betaine of histidine. A sulfur group is added to hercynine to form l-(+)-ergothioneine [90,94,95].

Mammals acquire l-(+)-ergothioneine solely through their diet: foods such as mushrooms, black beans, red meat, and oats are rich in l-(+)-ergothioneine. l-(+)-Ergothioneine is cumulated in cells and tissues frequently exposed to oxidative stress with the highest levels in the millimolar range occurring in blood, crystalline lenses, liver, bone marrow, and seminal fluid. In the bovine lens, l-(+)-ergothioneine concentrations about 7 mmol/L 10-times exceed those of glutathione, generally regarded as the most abundant endogenous antioxidant. In the bovine cornea, l-(+)-ergothioneine concentrations are 14-fold higher than those of glutathione, suggesting that it is the principal antioxidant in this tissue. Patients with rheumatoid arthritis accumulate l-(+)-ergothioneine in their synoviocytes. l-(+)-Ergothioneine, when ingested, can scavenge reactive oxygen and nitrogen species and protect cells from a variety of apoptotic insults. l-(+)-Ergothioneine inhibits tumor necrosis factor-α induced release of the inflammatory cytokine interleukin-8 in alveolar macrophages. The presence of a high affinity transporter OCTN1 of l-(+)-ergothioneine in conjunction with its non-random distribution in the body strongly implies the physiologic role of l-(+)-ergothioneine. l-(+)-Ergothioneine scavenges hydroxyl radicals as well as directly absorbs UV radiation. l-(+)-Ergothioneine has an absorption spectrum in the UV range similar to DNA with a molar extinction coefficient of 1.4 × 10^−4^ mol^−1^.cm^−1^, λ_max_ = 257 nm, suggesting that l-(+)-ergothioneine can act as a physiological UV filter. In vitro and cell culture studies have identified l-(+)-ergothioneine also as a scavenger of superoxide anion radicals.

l-(+)-Ergothioneine may provide more stable mode of cytoprotection and since it is not metabolized to any notable extent in mammalian tissues, the half-life of dietary l-(+)-ergothioneine is approximately one month. These properties suggest a role for l-(+)-ergothioneine as a bulwark, an ultimate defense for cells against oxidative damage. Thus, l-(+)-ergothioneine appears to be an important physiologic cytoprotectant, which probably merits its designation as a vitamin.

Next, the reason to use membrane compositions based on the two self-associating biopolymers chitosan and HMM HA could be advocated as follows: the incorporation of the molecules of the cytoprotectant l-(+)-ergothioneine into a two-component viscous solution composed of self-associating chitosan and HA results after drying in a formation of composite membranes ready-to-use to heal chronic wounds. In order to provide a desired shape and to strengthen the membrane, it is advantageous to impregnate a sparsely woven fabric/gauze or even the whole bandage with the three-component viscous solution. The cytoprotectant l-(+)-ergothioneine can be gradually released from such membranes during the extended period of time. Since after a certain time, l-(+)-ergothioneine from the membrane penetrates and is incorporated into the wound tissue, the used membrane can be readily removed and substituted by a new one. One of the especially important advantages of the suggested formulation is that the composite membrane containing the cytoprotectant l-(+)-ergothioneine can be preserved for unlimited time, because l-(+)-ergothioneine is not oxidized spontaneously. It is, therefore, possible to store sterilized membranes in a mildly wet state in an appropriate container and to apply them to a wound at any required time. To summarize, why and how to exploit l-(+)-ergothioneine as a smart-released cytoprotectant for treatment of difficult-to-heal chronic wounds of skin/tissues, we should state that l-(+)-ergothioneine: 1. bears in its molecule a positive charge, i.e., is a cation, 2. its molecules (at least their thiol fraction) are chemically reactive, 3. is transported directly into the site of inflammation or to a vicinity of stressed cells by the high affinity OCTN1 vehicle.

Low molar mass of l-(+)-ergothioneine, i.e., less than 230 g/mol, which is substantially lower than molar masses of any MTA, perfectly fulfils a request for a limit of molar mass of drugs circulating in the organism, i.e., up to 400 g/mol. The requirement 1, i.e., a molecule as a cation is fulfilled by the presence of the atom N^+^ in l-(+)-ergothioneine (see Scheme 6). Concerning the requirement 2, as evident from Scheme 6, the molecule of l-(+)-ergothioneine is a tautomer between thione and thiol forms. While thione is a stable unoxidizable (reserve) fraction, “thiol” is an active principle, which due to the functional group –SH is an effective donor of H^●^ radical, and just H^●^ radical is a unique sufficiently reactive radical, which is able to scavenge the most reactive ROS, namely the ^●^OH radical. Very interesting is requirement 3, i.e., targeting the stressed cells. To fulfill this requirement, it is necessary to have a priori an endogenous vehicle in the organism, which could anchor and transport the molecule into the site of inflammation or in the vicinity of stressed cells. Moreover, exclusively for l-(+)-ergothioneine molecules, the existence of such a vehicle was affirmed: it is OCTN1 [91]. Moreover, l-(+)-ergothioneine administered as a drug is transferred by OCTN1 to stressed cells [92].

## Data Availability

Not applicable.

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
