# Peer review of "Self-Associating Polymers Chitosan and Hyaluronan for Constructing Composite Membranes as Skin-Wound Dressings Carrying Therapeutics"

_molecules, 2021, doi:10.3390/molecules26092535_

Round 1

Reviewer 1 Report

In this manuscript, the author summarized the mutual impact of two biopolymers on skin wound healing as skin wound dressings and carriers of remedies. Chitosan, industrially acquired by the alkaline N-deacetylation of chitin, belongs to β-N-acetyl-glucosamine polymers. Another biopolymer is hyaluronan (HA). Chitosan, a biodegradable, non-toxic, bacteriostatic and fungistatic biopolymer, has numerous applications in medicine. HA, one of the major structural components of the extracellular matrix in vertebrate tissues, is broadly exploited in medicine as well. I think additional data is requiring for publication as a full paper and there are several points requiring further modification.

  1. The expression of Table 1 and 2 is not clear enough. The author did not show the reader what does the blank space represent in “Water solubility” and “References” part. Besides, the author did not show the difference between blank space and space marked “-” in “Active/medical principle, drug” part. Tables could be designed to be clearer and more entertaining.
  2. The figures serial number is confused, repeated in many places. The author needs to correct the writing and expression of serial number.
  3. I think it is better for author to design a figure to demonstrate the mutual impact of chitosan and HA on skin wound healing and the various types of composite membranes.
  4. The expression of the HA should be highly consistent in the whole Paper. It is not recommended to alternately use “hyaluronic acid” and “hyaluronan”.
  5. Before the word abbreviation is indicated, the abbreviation should not appear. And after the abbreviation is indicated, the word should not appear frequently. The author needs to carefully check and correct the abbreviation for HA.
  6. The writing format of the references is confused, the author should unify them according to the requirements.
  7. A large number of important expression in the paper was lacking of literature support, especially in “Pharmacokinetics” part. For example, the author mentioned several characteristics of drugs incorporated into TDDS (the last paragraph in page 2), however, the basis of statement is not shown to the reader, so it is impossible to verify whether the statement is correct. I suggest the author to read the followed articles and use them as references.

1) Research 2020, 2020, 3672120.

2) Carbohydrate Polymers, 2020, 241, 116364.

3) International Journal of Pharmaceutics, 2020, 578, 119127.

4) Research 2020, 2020, 2760594.

5) Advanced Science 2020, 7, 2000789.

Author Response

Dear reviewer,

Thank you for your comments and questions. Our answers to them are written in red.

Also the changes in the revised manuscript are written in red.

In this manuscript, the author summarized the mutual impact of two biopolymers on skin wound healing as skin wound dressings and carriers of remedies. Chitosan, industrially acquired by the alkaline N-deacetylation of chitin, belongs to β-N-acetyl-glucosamine polymers. Another biopolymer is hyaluronan (HA). Chitosan, a biodegradable, non-toxic, bacteriostatic and fungistatic biopolymer, has numerous applications in medicine. HA, one of the major structural components of the extracellular matrix in vertebrate tissues, is broadly exploited in medicine as well. I think additional data is requiring for publication as a full paper and there are several points requiring further modification.

1.The expression of Table 1 and 2 is not clear enough. The author did not show the reader what does the blank space represent in “Water solubility” and “References” part. Besides, the author did not show the difference between blank space and space marked “-” in “Active/medical principle, drug” part. Tables could be designed to be clearer and more entertaining.

Thank you for the comment. In the revised manuscript in Table 1 the missing solubilities was completed for antibiotics - water soluble; for fentanyl, ketorolac and lidocaine -sparingly soluble in water.

The missing references were added for rotigotine, selegiline, estradiol and testosterone. The space marked “-” means that there is no active/medical principle, drug. It is explained in the Note.

In Table 2 the space marked “-” means minus and that there is no active/medical principle, drug.

2. The figures serial number is confused, repeated in many places. The author needs to correct the writing and expression of serial number.

Thank you for the comment. In the revised manuscript the serial number of figures was alright. Thank you for the comment and we renumbered schemes since we deleted the word sketch.

3. I think it is better for author to design a figure to demonstrate the mutual impact of chitosan and HA on skin wound healing and the various types of composite membranes.

Thank you for the comment. Our communication relates only to one type of composite membranes: comprising chitosan and hyaluronan. The figure of the design of such a composite is clearly demonstrated in Figures 1 and 2.

4. The expression of the HA should be highly consistent in the whole Paper. It is not recommended to alternately use “hyaluronic acid” and “hyaluronan”.

Thank you for the comment. The word hyaluronic acid was removed and replaced with the word hyaluronan.

5. Before the word abbreviation is indicated, the abbreviation should not appear. And after the abbreviation is indicated, the word should not appear frequently. The author needs to carefully check and correct the abbreviation for HA.

Thank you for the comment. In the revised manuscript we checked and corrected the abbreviation for HA.

6.The writing format of the references is confused, the author should unify them according to the requirements.

Thank you for the comment. In the revised manuscript we made corrections in the format of the references.

7. A large number of important expression in the paper was lacking of literature support, especially in “Pharmacokinetics” part. For example, the author mentioned several characteristics of drugs incorporated into TDDS (the last paragraph in page 2), however, the basis of statement is not shown to the reader, so it is impossible to verify whether the statement is correct. I suggest the author to read the followed articles and use them as references.

Research 2020, 2020, 3672120.

Carbohydrate Polymers, 2020, 241, 116364.

International Journal of Pharmaceutics, 2020, 578, 119127.

Research 2020, 2020, 2760594.

Advanced Science 2020, 7, 2000789.

Thank you for the comment. In the revised manuscipt we completed the above mentioned references.

Zhang, X.; Chen, G.; Yu, Y.; Sun, L.; Zhao, Y. Bioinspired adhesive and antibacterial microneedles for versatile transdermal drug delivery. Research 2020, 2020, 3672120.
5. Graça, M. F. P.; Miguel, S. P.; Cabral, C. S. D.; Correia, I. J. Hyaluronic acid-Based wound dressings: A review. Carbohydr. Polym. 2020, 241, 116364.
6. Zhu, J.; Tang, X.; Jia, Y.; Ho, C. T.; Huang, Q. Applications and delivery mechanisms of hyaluronic acid used for topical/transdermal delivery - A review. Int. J. Pharm. 2020, 578, 119127.
7. Wang, F.; Zhang, X.; Chen, G.; Zhao, Y. Living bacterial microneedles for fungal infection treatment. Research 2020, 2020, 2760594.
8. Zhang, H.; Chen, G.; Yu, Y.; Guo, J.; Tan, Q.; Zhao, Y. Microfluidic printing of slippery textiles for medical drainage around wounds. Adv. Sci. 2020, 7(16), 2000789.

Reviewer 2 Report

The manuscript which is submitted for publication to Molecules is entitled: Self-associating polymers chitosan and hyaluronan for constructing composite membranes as skin-wound dressings carrying therapeutics. It is co-authored by K. Valachova & L. Soltes.

The work described in the manuscript is a review that describes the role that polysaccharides are playing, essentially in the field of Skin-Wound dressings. These two polysaccharides are chitosan and hyaluronic acid, respectively. While displaying a polymeric backbone of comparable length and conformational flexibility, they exhibit opposite charge distribution. These features provide the structural basis governing their interactions and the subsequent materials and their biological activities' properties. One of these features is the formation of composite membranes that can encapsulate bio-active molecules. The authors provide an interesting/educational introduction section. The manuscript's main scope is a coverage of the several applications and ends their presentation by some interesting discussion about membranes encapsulating resveratrol and ergothioneine. The manuscript is completed with an interesting addendum about ergothioneine.

The manuscript deserves publication in Molecules, once some minor modifications have been taken in to account.

Line 99. The wording alpha versus beta to describe polysaccharides is by far too simplistic and erroneous. This sentence could be deleted. Maybe a figure showing the structural similarities of the two polysaccharides would be more important.

Scheme 1. The scheme does not bring any pertinent information to the article. It could be deleted.

Line 109. The nomenclature for N-acetylglucosamine is GlcpNAc (NAG is an old nomenclature)

Line 111. The description of chitin is by far too simplistic. The crystalline structure of chitin is the result of a complex biosynthesis which is accompanied by a reorganization of the chains to form antiparallel arrangements in the case of chitin alpha or parallel in the case of chitin beta. These features explain their insolubility and different reactivities.

Line 118-119.  While presenting the obtention of chitosan from chitin, by deacetylation NaOH, the authors should mention the resulting pattern of deacetylation, which is a feature of paramount influence in the distribution of charges along the chains

Line 121. The authors may want to replace « mean » with average

Line 125 Rephrase the description of HA. Again, the use of NAG should be avoided. The very high molecular weight of HA is observed only in some instances.  In this part, it could be mentioned that HA is a member of the glycosaminoglycan family. The authors are invited to visit the recent issue published in Biomolecules on the multifaceted features of GAGs, as a source of information to present HA.

Line 141; The use of «we » does not fit with the writing of the paragraph

Line 153 idem (34) is not your work

Line 162-163 Could you elaborate. Does the process modify the charge distribution or is it related to the way « native chitosan » has been prepared?

Line 177. Could you define the term « ergo » (for the non-expert readers)?

Line 299.  The expression « glue-like » could be changed some something more scientific (viscous, gelatinous,….. ?) There exist more than 50 synonyms. !

Author Response

Dear reviewer,

Thank you for your comments and questions. Our answers to them are written in red.

Also the changes in the revised manuscript are written in red.

Comments and Suggestions for Authors

The manuscript which is submitted for publication to Molecules is entitled: Self-associating polymers chitosan and hyaluronan for constructing composite membranes as skin-wound dressings carrying therapeutics. It is co-authored by K. Valachova & L. Soltes.

The work described in the manuscript is a review that describes the role that polysaccharides are playing, essentially in the field of Skin-Wound dressings. These two polysaccharides are chitosan and hyaluronic acid, respectively. While displaying a polymeric backbone of comparable length and conformational flexibility, they exhibit opposite charge distribution. These features provide the structural basis governing their interactions and the subsequent materials and their biological activities' properties. One of these features is the formation of composite membranes that can encapsulate bio-active molecules. The authors provide an interesting/educational introduction section. The manuscript's main scope is a coverage of the several applications and ends their presentation by some interesting discussion about membranes encapsulating resveratrol and ergothioneine. The manuscript is completed with an interesting addendum about ergothioneine.

We thank the reviewer for the appreciation.

The manuscript deserves publication in Molecules, once some minor modifications have been taken in to account.

Line 99. The wording alpha versus beta to describe polysaccharides is by far too simplistic and erroneous. This sentence could be deleted. Maybe a figure showing the structural similarities of the two polysaccharides would be more important.

Thank you for your comment. In the revised manuscript the following sentence was deleted “Both β-polymers chitin and hyaluronan as well as cellulose provide structure, while α-polymers, e.g., glycogen provide food and energy.“

Scheme 1. The scheme does not bring any pertinent information to the article. It could be deleted.

We think that for readers this Scheme 1 can be important, therefore this scheme was not deleted.

Line 109. The nomenclature for N-acetylglucosamine is GlcpNAc (NAG is an old nomenclature)

Thank you for the comment. In the revised manuscipt we replaced NAG with GlcpNAc.

Line 111. The description of chitin is by far too simplistic. The crystalline structure of chitin is the result of a complex biosynthesis which is accompanied by a reorganization of the chains to form antiparallel arrangements in the case of chitin alpha or parallel in the case of chitin beta. These features explain their insolubility and different reactivities.

Thank you for the comment. In the revised manuscipt the sentence “Prolongation of the molecular size to oligomeric or polymeric one due to increased hydrogen bonding between adjacent polymer chains resulted in the formation of chitin-polymer matrix with increased strength and water insolubility“ was replaced with “The crystalline structure of chitin is the result of a complex biosynthesis which is accompanied by a reorganization of the chains to form antiparallel arrangements in the case of chitin alpha or parallel in the case of chitin beta. These features explain their insolubility and different reactivities“.

Line 118-119.  While presenting the obtention of chitosan from chitin, by deacetylation NaOH, the authors should mention the resulting pattern of deacetylation, which is a feature of paramount influence in the distribution of charges along the chains.

Thank you for the comment. Chitin is not charged, however chitosan is charged. We did not understand what is “the obtention“ of chitosan from chitin.

Line 121. The authors may want to replace « mean » with average

In the revised manuscipt the word mean was replaced with the word average.

Line 125. Rephrase the description of HA. Again, the use of NAG should be avoided. The very high molecular weight of HA is observed only in some instances.  In this part, it could be mentioned that HA is a member of the glycosaminoglycan family. The authors are invited to visit the recent issue published in Biomolecules on the multifaceted features of GAGs, as a source of information to present HA.

Thank you for your comment. In the revised manuscipt the sentence was rewritten as follows: “Hyaluronan (HA), a biopolymer, is composed of repeating disaccharide units of GlcpNAc and d-glucuronic acid, linked exclusively by β-linkages. HA is a member of the glycosaminoglycan family [Valachová, K.; Šoltés, L. Versatile use of chitosan and hyaluronan in medicine. Molecules 2021, 26, 1195; Sodhi, H.; Panitch, A. Glycosaminoglycans in tissue engineering: A review. Biomolecules 2021, 11(1), 29.]“.

Line 141. The use of «we » does not fit with the writing of the paragraph

We agree. In the revised manuscipt the sentence “After the addition of another component such as an “active principle” to the pre-formulated viscous solution of those two polymers we prepare a composite membrane, which is ready-to-use for treatment of, e.g., difficult-to-heal chronic skin wounds“ was changed to “After the addition of another component such as an “active principle” to the pre-formulated viscous solution of those two polymers a composite membrane was prepared, which is ready-to-use for treatment of, e.g., difficult-to-heal chronic skin wounds.“

Line 153, item (34) is not your work

The reference [34] is related to Behin, S.R.; Punitha, I.S.; Saju, F. Development of matrix dispersion transdermal therapeutic system containing glipizide. Der Pharm. Lett. 2013, 5, 278−286.

Line 162-163 Could you elaborate. Does the process modify the charge distribution or is it related to the way « native chitosan » has been prepared?

In the revised manuscript the following sentence was deleted. “Yet, it is necessary to point out here that the distribution of charges along the macromo-lecular chains of chitosan and its derivatives differs significantly“.

Line 177. Could you define the term « ergo » (for the non-expert readers)?

Thank you for the question. In the revised manuscript the term « ergo » was deleted and replaced with the word thus.

Line 299.  The expression « glue-like » could be changed some something more scientific (viscous, gelatinous,….. ?) There exist more than 50 synonyms. !

Thank you for your comment. In the revised manuscipt the term « glue-like » was replaced with the word viscous.

Round 2

Reviewer 1 Report

The paper has been well improved. Most of my concerns are addressed with additional discussions concerning several obscures provided. Some details are also included in the revised manuscript. I am therefore pleased to see it published as its current form.